# Mitochondrial Epilepsy, a Challenge for Neurologists

**DOI:** 10.3390/ijms232113216

**Published:** 2022-10-30

**Authors:** Piervito Lopriore, Fábio Gomes, Vincenzo Montano, Gabriele Siciliano, Michelangelo Mancuso

**Affiliations:** 1Neurological Institute, Department of Clinical and Experimental Medicine, University of Pisa, 56126 Pisa, Italy; 2Neurology Department, Coimbra University Hospital Centre, 3004-561 Coimbra, Portugal

**Keywords:** primary mitochondrial disease, epilepsy, mitochondrial epilepsy, MELAS, MERRF, *POLG*-related disorders, Leigh syndrome, stroke-like episode

## Abstract

Primary mitochondrial diseases are relatively common inborn errors of energy metabolism, with a combined prevalence of 1 in 4300. These disorders typically affect tissues with high energy requirements, including the brain. Epilepsy affects >1% of the worldwide population, making it one of the most common neurological illnesses; it may be the presenting feature of a mitochondrial disease, but is often part of a multisystem clinical presentation. The major genetic causes of mitochondrial epilepsy are mutations in mitochondrial DNA and in the nuclear-encoded gene *POLG*. Treatment of mitochondrial epilepsy may be challenging, often representing a poor prognostic feature. This narrative review will cover the most recent advances in the field of mitochondrial epilepsy, from pathophysiology and genetic etiologies to phenotype and treatment options.

## 1. Introduction

Mitochondria are complex cytoplasmic, double-membrane organelles, thought to have originated from endosymbiosis of a primordial eukaryote with free-living eubacteria. For decades, they have only been considered the power stations of eukaryotic cells, orchestrating cellular energy production in the form of ATP molecules through the activity of their respiratory chains. Beyond ATP production, mitochondria are central metabolic gatekeepers, maintaining cellular ion homeostasis, producing precursors for macromolecules and sequestering metabolic catabolites [1]. 

However, mitochondria influence a diversity of cellular systems beyond bioenergetics, playing active roles as intracellular signaling platforms, regulating cellular response to stressors, cell growth, differentiation and death. Thus, mitochondrial dysfunction has emerged as a key factor in the pathogenesis of metabolic diseases, cancer growth and neurodegeneration. Moreover, disrupted mitochondrial function is a hallmark of aging [1].

Primary mitochondrial diseases are the most frequent metabolic disorders in humans, with a prevalence of approximately 1 in 4300 cases [2]. They may result both from pathogenic variants in all 37 mitochondrial DNA (mtDNA) genes and in more than 400 nuclear DNA (nDNA) genes [3]. The unique features of mitochondrial genetics explain the vast clinical heterogeneity of primary mitochondrial diseases, which are characterized by direct or indirect defects of the mitochondrial respiratory chain, the site of oxidative phosphorylation (OXPHOS) [1]. Since mitochondria are essential organelles in all human cells, mitochondrial diseases can affect virtually all organs, with a predilection for high-energy demanding tissues, such as the muscle, central and peripheral nervous systems, heart and retina [4]. Primary mitochondrial diseases may manifest in childhood or adulthood, with symptoms affecting a single organ or multiple organ or systems. Phenotypic expression varies depending on age at onset. Adults usually manifest signs of myopathy in association with variable peripheral and central nervous system involvements. Infants frequently exhibit psychomotor delay, hypotonia, lactic acidosis and cardiorespiratory failure in the context of complex multisystem disorders, severe encephalomyopathies or isolated myopathies accompanied by cardiomyopathies. 

In both infants and adults, epilepsy is a major feature of primary mitochondrial diseases. It has pleiomorphic characteristics and can occur sporadically, but it is often part of complex and specific phenotypes. Studies on mitochondrial epilepsy pathomechanisms offer evidence of a cardinal role of mitochondrial metabolism in seizure generation. Moreover, novel genetic analysis, such as multi-omics, provide better knowledge of the molecular genetics behind mitochondrial epilepsy, increasing the number of genes responsible for this phenotype. Improving our knowledge of epilepsy in primary mitochondrial diseases will allow us to better diagnose these disorders and gives us hope for the development of personalized treatments for patients. 

## 2. Epilepsy in Primary Mitochondrial Disease

Epilepsy is the most common feature of central nervous system involvement in primary mitochondrial diseases. Approximately 20–50% of patients will have seizures during their disease course [5]. In nearly 20% of early-onset primary mitochondrial diseases, seizures may represent a presenting symptom, and globally, they have been reported to occur in 35–60% of pediatric patients [6]. One prospective United Kingdom (UK) study on a cohort of adult patients, as well as a large retrospective adult-prevalent study from Italy, found an overall prevalence of epilepsy of 10–23% [7,8]. Seizures in primary mitochondrial diseases recur in approximately 90% of cases; patients with early-onset epilepsies have a higher frequency of seizures compared with individuals with adult-onset [8]. Interestingly, in the UK cohort, epilepsy did not itself appear to contribute to increased mortality [7].

In primary mitochondrial diseases, any seizure type can occur; the most common are myoclonus, focal motor seizures with secondary generalization and generalized tonic-clonic seizures, the rarest classical absence seizures [5]. Seizure semeiology varies with the age of onset: patients with early-onset epilepsy have a higher probability of displaying seizures of unknown cause as the first seizure type, as well as spasms and tonic seizures during the disease course; patients with adult onset have a higher probability of myoclonus as the first type of generalized seizure [8]. Mitochondrial patients may experience status epilepticus, either non-convulsive or focal motor (also called epilepsia partialis continua). The prevalence of status epilepticus in primary mitochondrial diseases is unknown, but it usually co-exists with stroke-like episodes (SLEs). Status epilepticus in primary mitochondrial diseases has a poor outcome with a high mortality rate, given its treatment resistance [9]. Remarkably, combinations of different seizure types are present in 20–60% of mitochondrial patients [5,7,8]. Complex epilepsy syndromes, presenting as progressive early-onset epileptic encephalopathy (such as Lennox–Gastaut, West or Othahara syndromes) may also occur [6,10].

Epilepsy in primary mitochondrial diseases does not have pathognomonic electroencephalography (EEG) findings. In approximately 90% of patients, EEG traces display a specific abnormal background activity, consisting of diffuse or focal slowing or interictal paroxysm [8]. These findings are more common in individuals with early-onset epilepsy. In 10% of mitochondrial patients, especially in those with occasional seizures, an interictal EEG trace is normal. Ictal EEG typically presents focal spikes or multifocal spikes and, sometimes, generalized spike-wave activity or hypsarrhythmia [5,7,8]. Epileptiform abnormalities most commonly localize to the posterior temporal, parietal and occipital regions [8,11]. Interestingly, brain MRI abnormalities, mostly T2 hyperintensities and abnormal diffusion restriction, frequently involve the parieto-occipital lobes, often in concordance with EEG findings [11].

Epilepsy can be observed in primary mitochondrial diseases caused by both nDNA and mtDNA mutations [5,8]. Several primary mitochondrial disease phenotypes can present epilepsy. In Table 1 we summarized the gene defects and phenotypes associated with mitochondrial epilepsy. Generally, the relationship between genotype and phenotype is not absolute, and there is a significant overlap between syndromes [12]. Mitochondrial Encephalopathy, Lactic Acidosis, Stroke-like episodes (MELAS), Myoclonic epilepsy with ragged red fibers (MERRF), *POLG*-related disorders and Leigh syndrome (LS) are the most common primary mitochondrial diseases in which epilepsy can occur and will be the focus of this review. Other phenotypes, such as pyruvate dehydrogenase complex deficiency or other mitochondrial encephalopathy, may present epilepsy as a clinical feature. Why some gene defects predispose patients to the development of epilepsy is not fully understood. A biochemical explanation seems partial, and additional factors, such as nuclear polymorphism and heteroplasmy level, may have contributing roles [5,13]. At the same time, genetics influence disease severity; it has been observed that individuals with early-onset and severe epilepsy phenotype most commonly carry nDNA pathogenic variants [8].

## 3. Pathomechanism

The International League Against Epilepsy (ILAE) defines an epileptic seizure as “a transient occurrence of signs and/or symptoms due to abnormal excessive or synchronous neuronal activity in the brain” [5]. The transient and superabundant burst of neuronal activity results from uncontrolled membrane depolarization caused by altered ionic or synaptic transmission. An imbalance between stimulatory and inhibitory neurotransmitters is thought to cause the spreading of this hyperexcitability through the cortex. However, the impaired interplay between glutamate-mediated excitation and GABA-mediated inhibition is just a frame of the complex picture; epileptogenesis can involve other biological pathways responsible for structural and functional changes within the brain, such as channelopathies, aberrant hippocampal neurogenesis and immune system and inflammatory reactions [6].

Neurons rely mostly on aerobic metabolism to maintain membrane polarization through the action of Na^+^ and Ca^++^ channels. Recent findings have shown that there is a significant association between epilepsy and metabolism within the brain [8]. Considering this, the ATP depletion resulting from defective OXPHOS mainly explains the occurrence of seizures in primary mitochondrial diseases (Figure 1). On one hand, impaired membrane channels and Na^+^/K^+^ ATPase activity results in the loss of neuron hyperpolarization. On the other hand, defective mitochondrial metabolism leads to excessive cell excitation through diverse mechanisms: (1) lowering of GABA-mediated inhibition resulted from death of inhibitory interneurons (especially in the occipital cortex); (2) suppression of hippocampal intermediate inhibitory neurons activity; (3) increasing of glutamate release in the synaptic space by astrocytes. Notably, interneurons appear to be more susceptible than primary neurons to the OXPHOS deficiencies, especially those resulting from complex I and complex IV defects [8,9,10]. This can explain why not all complexes’ defects provoke epilepsy with the same frequency and degree, and why some genetic defects are more prone to causing epilepsy (Table 1). An astrocyte–neuron metabolic interplay through the action of a lactate shuttle is supposed to primarily sustain the energetic requirement of neurons, especially during high synaptic activity [11].

Recently, some evidence on rodent brain slice models supported the ‘dual neuronal–astrocytic hypotheses’ of mitochondrial epilepsy generation. OXPHOS deficiency, together with the downregulation of glutamine synthetase, reduced recycling of GABA in astrocytes. Astrocytic metabolic shutdown complements inhibitory interneuron death in the generation of a hyperexcitable network, which supports seizure generation [13]. 

Additionally, ATP depletion may lead to damage of brain structures, which, in turn, may create epileptogenic foci; exemplary are the cases of neonatal encephalopathy forms of pyruvate dehydrogenase complex deficits, which cause intrauterine white matter loss and SLEs in MELAS or *POLG*-diseases, which may evolve into encephalomalacia or glial scars [9,10]. Because mitochondria store intracellular calcium, there is evidence that mitochondrial dysfunction can impair the calcium cycle, exposing neurons to damage [14]. Cellular calcium overload and OXPHOS dysfunction together are responsible for the generation of an excessive amount of reactive oxygen/nitrogen species (ROS and RNS), causing oxidative damage in the form of reactive gliosis, lipid peroxidation and mtDNA damage; this results in neurodegeneration, re-arrangement of neuronal circuits and neuronal hyperexcitability and a lowered seizure threshold [8,15,16]. 

Notably, seizures themselves can trigger mitochondrial dysfunction, generating a self-perpetuating cycle. During severe episodes of seizure exacerbation, stroke-like lesions and status epilepticus may develop [8,17]. Thus, even though the underlying mechanism of SLEs remains unclear, they are generally regarded as manifestation of prolonged and aberrant ictal activity [18]. 

Interestingly, immune dysfunction has been proposed as a possible actor in mitochondrial epilepsy pathophysiology. There is evidence of blood–brain barrier disruption and autoantibodies detection in the blood and CSF of epileptic patients harboring *POLG* mutations [19,20]. Finally, electrolyte disturbances caused by mitochondrial tubulopathy may secondarily cause seizures.

## 4. Mitochondrial Epilepsy Phenotypes

### 4.1. Mitochondrial Encephalopathy, Lactic Acidosis and Stroke-like Episodes—MELAS

MELAS syndrome is a multi-organ disease characterized by encephalopathy that manifests through seizures and/or dementia, mitochondrial myopathy with lactic acidosis and stroke-like episodes. The disease typically occurs before age of 40 [5,57]. MELAS has a worldwide distribution, without any known ethnic predilection. According to epidemiological studies, its prevalence ranges from at least 18.4:100,000 to 236:100,000 [57]. 

This syndrome is caused by a vast number of mutations in the mtDNA, with the m.3243A > G point mutation in the MT-TL1 gene encoding the mitochondrial tRNA (Leu(UUR)) being the most common (approximately 80% of patients). The m.13513G > A mutation in the gene encoding subunit 5 of complex I (MT-ND5) represents the second-most common genetic cause [5,12].

In two cohort studies of patients with mitochondrial disease and epilepsy—one from the UK and one from Italy—the m.3243A > G point mutation was the most common genotype found [7,8]. In the UK cohort, 34.9% of patients who harbored this mutation developed epilepsy. In the Italian cohort, 41.8% of the patients of the subgroup with epilepsy carried it [7]. Additionally, another work observed that the male gender could represent a risk factor for the development of SLEs in m.3243A > G patients [58].

Epilepsy in MELAS is heterogeneous. In a series of 34 patients from China, the age of onset of the epileptic seizures ranged from 0.5 to 57 years, with an average of 22.6; focal onset seizures (clonic and tonic) were the most reported, followed by generalized onset, tonic–clonic seizures [59]. This variability was confirmed in other studies, including ones focusing on the pediatric population [59,60]. Seizures can be either spontaneous or provoked, attesting to the metabolic nature of the disease [59]. Focal seizures can manifest in the context of an acute SLE or even a sequela of prior episodes. Generalized seizures may also occur during SLEs and in the context of structural abnormalities [57]. Although less commonly, myoclonic seizures have been described in MELAS patients [5].

Status epilepticus can occur in MELAS. In one series of patients, all cases of status epilepticus (5 out of 63 patients) were associated with SLEs [7]. Moreover, the occurrence of status epilepticus has been reported in the context of SLE as the first manifestation of the disease [9]. Both convulsive and nonconvulsive status epilepticus have been described, with some reports of the latter manifesting through behavioral changes and psychosis [61]. Globally, status epilepticus correlates with poor prognosis [59].

SLE is the clinical hallmark of MELAS, usually manifesting before the age of 40 and occurring in 84–99% of individuals [14,57]. Although, in the acute phase, an SLE may mimic an acute stroke—with focal neurological deficits such as reversible aphasia, motor deficit or cortical blindness—there is a growing acceptance that they represent seizure activity [14,62]. These episodes are conceptualized as a form of focal status epilepticus with secondary encephalopathy. In fact, some individuals will only have seizures in the context of SLEs.

SLEs are accompanied by correspondent stroke-like lesions in the brain MRI that do not conform to typical vascular territories [14]. Stroke-like lesions usually affect the cortical and subcortical white matter of the posterior brain. However, the thalamus and other deep brain grey matter structures can be affected as well [63]. In the acute phase, stroke-like lesions present as hyperintensities on the MRI T2-weighted scans, with cortical lesions usually showing patchy or linear enhancement on T1-weighted postcontrast images. Stroke-like lesions usually present a high apparent diffusion coefficient, suggesting the presence of vasogenic edema, differently to what is seen in acute stroke. Nonetheless, a decrease in the apparent diffusion coefficient compatible with cytotoxic oedema has been reported in some cases as well [57]. These alterations change during the sub-acute and chronic phase of SLEs, and stroke-like lesions usually migrate to adjacent areas of the cortex over time [63]. Lactate peak and decreased N-acetyl aspartate concentration in spectroscopy may be present before the occurrence of the aforementioned SLE changes. Nonetheless, it is known that this peak may also be present in normal-appearing brain areas, and that the lactate level fluctuates during the disease course [63]. Angiography sequence studies may reveal major vessel dilatations which correlate with the occurrence of SLEs. These findings illustrate the potential role of different brain MRI modalities in the risk stratification and monitoring of MELAS patients [63].

### 4.2. Myoclonic Epilepsy with Ragged Red Fibers—MERRF

MERRF is a maternally inherited mitochondrial encephalomyopathy. It usually occurs in childhood or early adulthood with a slowly progressive course, whereas adolescents may develop it rapidly, with a fatal outcome. Overall, age of death has ranged from 7 to 79 years of age. In addition, 80% of patients have a positive family history; even so, phenotypes vary significantly among family members. This can be partially explained by the difference in the heteroplasmy of mutated mtDNA [15].

More than 75% of MERRF patients harbor the m.8344A > G mutations in the MT-TK gene encoding the mitochondrial tRNA for lysine. In the adult population, the prevalence of this point mutation varies from 0.39:100,000 to 1.5:100,000 [1,9]. Two other mutations in the same gene (m.8356T > C and m.8361G > A) have been recognized as pathogenic [5].

Myopathy (with ragged-red fibers) and progressive myoclonic epilepsy are the crucial features of the syndrome. Myoclonus can be either continuous or intermittent; it is usually photosensitive and aggravated by stimuli and actions such as writing or eating. It is important to underline that myoclonus in MERRF may also result from medullo-cerebellar dysfunction [5]. Additionally, most individuals present generalized tonic–clonic seizures. Some patients experience atonic and absence seizures, and there are reports of focal seizures as well [5]. In one Italian cohort study, generalized epilepsy was common to all deceased individuals, thus representing a putative negative prognostic factor [15]. 

Previous studies have dissected the phenotypic variability of the m.8344A > G mtDNA mutation. The full-blown phenotype can include cerebellar ataxia, dementia, sensorineural hearing loss and multiple cutaneous lipomas, among other features. In fact, it has been shown that the presence of myoclonus is more commonly associated with ataxia than with generalized seizures, sustaining the idea of a pivotal role of the cerebellum in the development of cortical myoclonus [15]. Status epilepticus may occur in MERRF patients and, as for MELAS, it can be associated with SLEs [9]. Furthermore, as already mentioned, overlapping syndromes between MERRF and other phenotypes, such as MELAS and Kearns Sayre syndrome, have been reported [5].

The brain MRI may be normal, or it may show basal ganglia calcification, cerebral, cerebellar and/or brainstem atrophy, white matter abnormalities, cysts or vacuolated lesions and stroke-like lesions. Neuronal loss and gliosis in the brain involve, preferentially, the cerebellar dentate nucleus, the inferior olivary nucleus, the red nucleus and the substantia nigra, as well as the thoracic nucleus of Clarke and the spinal anterior and posterior horns. Demyelination preferentially affects the superior cerebellar peduncles as well as the posterior columns and lateral spinocerebellar tracts of the spinal cord, whereas the pyramidal system is usually spared or mildly affected [15]. In EEG, myoclonus can translate into spikes and polyspikes [5].

### 4.3. POLG-Related Disorders

Mutations in the nuclear-encoded POLG gene lead to a group of heterogeneous, frequently overlapping phenotypes that differ in severity and age of onset [5,64]. POLG mutations represent the most common cause of mitochondrial epilepsy at all ages. In a UK epidemiological study, the prevalence of autosomal recessive POLG mutations with clinical manifestations was 0.3:100,000 adults [17]. 

POLG is a nuclear gene which encodes the polymerase gamma, a DNA polymerase responsible for mtDNA replication and repair [10]. POLG mutations may lead to mtDNA point mutations, multiple deletions or quantitative depletion, and are the most common cause of primary mitochondrial diseases [64]. The three most frequent pathogenic variants are c.1399G > A (p.Ala467Thr), c.2243G > C (p.Trp748Ser) and c.2542G > A (p.Gly848Ser) [64]. To date, more than 190 disease-causing variants of the POLG gene have been identified, and about 100 of them have been linked to epilepsy. Epilepsy is a frequent comorbidity in POLG-related disorders, especially in childhood phenotypes, globally affecting 50–65% of individuals [64,65]. 

Phenotypes developing during childhood are frequently the result of mtDNA depletion, and these include Alpers–Huttenlocher syndrome and myocerebrohepatopathy spectrum [65] (Table 1). Alpers–Huttenlocher syndrome is hallmarked by refractory seizures, neurodevelopmental regression and liver dysfunction (developing later in the disease course) [66]. Alpers–Huttenlocher syndrome is responsible for the majority of early-onset epilepsy cases and represents the most frequent cause of mitochondrial status epilepticus [9,64]. Most children live a few months after the onset; some patients may survive longer, developing myoclonic seizures and refractory status epilepticus as the disease progresses [5,64]. 

POLG-related phenotypes with juvenile/adult onset are typically characterized by multiple mtDNA deletions and tend to develop phenotypes included in the myoclonic epilepsy myopathy sensory ataxia and ataxia neuropathy spectrum. MEMSA covers the disorders previously described as spinocerebellar ataxia with epilepsy. 

The MELAS phenotype may also develop with certain POLG mutations [17,67]. In these groups of patients, seizures are present in 75% of cases, and occipital lobe seizures are common and frequently associated with headache and vomiting [5,64]. Almost all patients also present with focal clonic and myoclonic seizures, and progression to status epilepticus is common [64,66].

In a multinational cohort, 68% of children with early-onset POLG disease developed status epilepticus and 58% presented epilepsia partialis continua during the disease course [9]. The most common seizure type for both adult and pediatric populations is focal seizures, which frequently generalize to bilateral convulsive seizures [64]. Epilepsy has been associated with a poor prognosis in these individuals [65].

A systematic review of neuroimaging in POLG-related epilepsy showed that stroke-like lesions were the most reported abnormalities [5,17]. Cortical lesions may develop as result of status epilepticus and prolonged seizure activity, with T2 hyperintensities that may regress after weeks. Spectroscopy analysis on brain MRIs follows the features seen in MELAS patients. EEG usually displays an interictal epileptic activity that correlates anatomically with cortical lesions during exacerbations [64]. EEG in Alpers–Huttenlocher syndrome patients may show occipital rhythmic high-amplitude delta with superimposed polyspikes (RHADS) [68]. Recent studies have shown that POLG mutations are associated with progressive disruption of the blood–brain barrier. Elevated CSF protein correlates with early-onset severe phenotypes and epilepsy in POLG patients, making the CSF protein level a potential biomarker of disease severity and prognosis [65].

### 4.4. Leigh Syndrome—LS

LS is a subacute necrotizing encephalomyelopathy, which usually manifests in infancy or early childhood with phenotypic and genetic heterogeneity [6]. It is marked by psychomotor regression or development delay, bilateral specific changes of the basal ganglia and brainstem and altered mitochondrial metabolism. LS prevalence is estimated to be at least 1 in 40,000 newborns, with some populations around the world presenting a higher number of cases [69]. 

LS can be caused by both mtDNA (maternal inherited LS or MILS) and nDNA. More than 75 nuclear disease-causing monogenic mutations have been recognized. Most of these genes encode structural components of the OXPHOS complexes [69]. Complex I deficiency is the most common biochemical cause of LS, particularly in individuals with mutations of the ND5 subunit [6,69]. Among primary mitochondrial disease phenotypes, LS is the most frequently associated with complex IV deficiency; however, seizures are a rare feature for most of these patients. Mutations in the mtDNA-encoded ATP6 gene, encoding for a complex V subunit, are a common cause of MILS [6]. However, LS can also be caused by mutations in nuclear genes encoding proteins required for complex assembly, stability and activity [69].

Seizures are present in around 40% of LS patients, and different types of seizures may occur [6]. Myoclonic, focal and generalized tonic–clonic seizures have been described so far [6,66]. MILS may present with infantile spasms that usually respond well to pharmacological treatment [6]. Nevertheless, most of the patients will develop drug resistant epilepsy.

The development of bilateral and symmetrical lesions, hyperintense in T2, of the brainstem and basal ganglia represent the hallmark of LS, even though there are reports of LS patients who did not manifest these alterations. Cerebral atrophy and white matter lesions may be found as well [20,69].

## 5. Management

### 5.1. Pharmacological Treatment

For most of primary mitochondrial diseases, there are no disease-modifying therapies; however, there are some exceptions that must be recognized early on. The most notable ones are the disorders of coenzyme Q10 (ubiquinone) biosynthesis, which may partially respond favorably with high-dose supplementation. Moreover, patients harboring mutations in the *ACAD9* gene leading to complex I deficiencies may present seizures and respond to riboflavin (vitamin B2) supplementation [9,62]. LS or Leigh-like syndromes caused by biotinidase deficiency or thiamine transporter 2 deficiency may respond to biotin (Vitamin B8) or thiamine (Vitamin B1) administration, respectively [66].

In 2017, a consensus-based statement from the Mitochondrial Medicine Society highlighted the importance of having a low threshold for obtaining an EEG registration in patients who present with any alterations to their previous cognitive state and repetitive stereotypical spells. The document states that no combination of anti-seizure medication has been proven to be superior [70]. An international Delphi-based consensus concerning the safety of drug use in patients with primary mitochondrial disease, including anti-epileptic drugs, was published in 2020 [71]. Overall, epilepsy in primary mitochondrial diseases should be treated similarly as in non-mitochondrial epilepsy. However, it is important to avoid certain drugs with known mitochondrial toxicity, such as sodium valproate, in patients with *POLG* mutations (due to the risk of hepatic failure and seizure aggravation) [70].

Currently, levetiracetam is regarded as a safe treatment, and it is a first choice for myoclonus, especially in MERRF. It can be combined with benzodiazepines (such as clonazepam or clobazam), which have been reported to be effective [6,68]. Lamotrigine is safe and effective, but it may worsen myoclonic seizures. Some reports indicate zonisamide as a safe option and lacosamide as a good adjunctive treatment in patients with MELAS or drug resistant seizures [64,68,70,72]. Gabapentin, oxcarbazepine, rufinamide and stiripentol are safe options as well [68]. There are reports of mitochondrial patients treated with oxcarbazepine, phenytoin or phenobarbital; nevertheless, the toxicity of these drugs and the potential negative effect on myoclonus should prompt clinicians to use other pharmacological options [64,68]. Finally, most patients need multiple anti-seizure drugs to achieve seizure control, and many individuals with mitochondrial epilepsy become resistant to therapy [62,68]. Treatment of status epilepticus and SLEs will be discussed later.

### 5.2. Non-Pharmacological Treatment

Since many patients develop drug resistant epilepsy, other therapeutic alternatives are often used. There are some reports regarding vagal nerve stimulation (VNS), deep brain stimulation (DBS) and palliative surgery use. Unclear results in seizure control have been obtained with VNS [10,73]. In 10 pediatric patients with drug resistant epilepsy, corpus callosotomy managed to reduce the frequency of seizures [74].

The ketogenic diet has been widely used in patients with mitochondrial intractable epilepsy [6,75]. A low-carbohydrate, high-fat diet stimulates fatty-acid utilization by beta-oxidation, producing ketone bodies; this process has beneficial results through stimulation of mitochondrial biogenesis and OXPHOS functioning as well as a reduction in oxidative stress. A ketogenic diet leads to a reduction in the glutamate level in the synaptic space, and increases levels of decanoic acid, a fatty acid that has been shown to reduce neuronal excitation. Furthermore, ketone bodies’ metabolism supports ATP production, partially bypassing complex I activity; thus, patients with complex I deficiencies are potential candidates for ketogenic diet treatment [6,68,75]. Interestingly, some studies have shown greater effectiveness of the ketogenic diet in patients harboring heteroplasmic mtDNA multiple deletions [6]. 

In a recent systematic review, the ketogenic diet showed a high efficacy in seizure control in mitochondrial patients, mostly pediatric; 65% of the cases presented with adverse effects such as headache, lethargy, rhabdomyolysis and lactic acidosis. Recently, a prospective open-labeled controlled study from China demonstrated that 31.8% of participants achieved ≥50% seizure reduction after 1 month of a ketogenic diet, which increased to 40.9% at 3 months, in a significant way when compared to the control group. Response rates were higher in patients with MELAS or pathogenic variants in mtDNA. In conclusion, the ketogenic diet has been proven to be safe and effective for seizure control in primary mitochondrial diseases, especially MELAS and mtDNA pathogenic variants [76]. Interestingly, the ketogenic diet seems to be contraindicated for treatment of mitochondrial myopathy related to mtDNA multiple deletions. Although data, so far, are too scarce for general guidelines, it is still recommended to consider a ketogenic diet in patients with resistant mitochondrial epilepsy, initiated by an experienced team [75,76].

### 5.3. Stroke like Episodes and Status Epilepticus 

In 2019, a consensus-based statement on SLE management stated that patients with previous episodes and presenting with suggestive signs must be treated quickly with benzodiazepines [77]. Once in the emergency department, intravenous anti-seizure therapy should be immediately initiated. Levetiracetam is recommended (20–40 mg/kg, max 4500 mg) but phenytoin, phenobarbitone or lacosamide can also be used [77]. The evidence for the use of L-arginine is scarce, and, therefore, is not recommended in the treatment of SLEs [8,77]. A prospective randomized controlled trial is needed to evaluate any compounds to treat or prevent SLEs.

As we already said, status epilepticus is generally associated with SLEs. If needed, midazolam should be the first option for general anesthetics when treating refractory status epilepticus in primary mitochondrial diseases, and these patients should be under continuous EEG monitoring and treated quickly and aggressively [63]. Propofol is not contraindicated. In case of convulsive status epilepticus, it is recommended to follow the local status epilepticus guidelines [77]. There are reports of positive response with several alternative interventions, such as the use of perampanel, corticosteroids, ketamine, immunoglobulin and magnesium infusion [9,64,78]. 

### 5.4. New Therapeutic Approaches

A considerable number of new therapeutic options for primary mitochondrial diseases are currently being studied. These include pharmacologic alternatives using antioxidant approaches, targeting mitophagy and mitochondrial biogenesis or stabilizing the mitochondrial membrane [62]. 

In genetic therapies, targeting the mtDNA genome is a difficult approach, since it is encased by two lipidic membranes. Good results have been shown in cellular and animal models of mtDNA-related disorders using specific nucleases to target and eliminate mutant mtDNA molecules [79].

Preclinical studies with adeno-associated viral vector-mediated gene therapy targeting the nuclear genome in primary mitochondrial diseases showed positive results. Nevertheless, the costs associated with developing models for hundreds of nuclear mutations make genetic therapies an unfeasible option for most patients in the near future [62].

At the time of writing (28 September 2022), a search for ‘mitochondrial diseases’ identified 29 interventional studies (clinical trials) in clinicaltrials.gov.

Currently, a double-blind placebo-controlled study to evaluate efficacy and safety of Vatiquinone for treating refractory epilepsy in primary mitochondrial diseases, including MERRF, is running (NCT04378075) [79].

## 6. How to Recognize a Mitochondrial Epilepsy?

As we discussed in this review, epilepsy manifestation in primary mitochondrial diseases is heterogenous; the common assumption that myoclonic seizures or severe epileptic encephalopathy should raise the suspicion of a mitochondrial disorder is incorrect. Thus, recognizing a mitochondrial epilepsy is a challenge, even for experienced neurologists. 

In mitochondrial medicine, family history is the first point to address; it must be taken meticulously, with special attention to minimal and apparently unspecific signs in the family (“mitochondrial red flags”), including short stature, diabetes, migraine, hearing loss, ataxia, epilepsy, exercise intolerance, cardiomyopathies and psychiatric disorders. A full list of the clinical features of primary mitochondrial diseases is shown in Figure 2. Clearly, a positive family history in the maternal lineage suggests a mtDNA-related disorder. Irrespective of family history, a primary mitochondrial disease should be suspected in patients with an apparently unrelated involvement of two or more tissues [80].

Seizures may be the presenting feature of primary mitochondrial diseases, especially in pediatric patient harboring nDNA mutations. Nevertheless, they are often part of a multisystem presentation and can occur during the disease course [5,8]. Mitochondrial epilepsy is often multifocal, showing both focal and generalized features. Clear syndromic epilepsy can be found in specific phenotypes, as discussed in the “Mitochondrial epilepsy phenotypes” paragraph. Status epilepticus, especially in the form of epilepsia partialis continua, when present, should raise the suspicion of a SLE, which is commonly associated with MELAS syndrome or *POLG*-related disorders [5,80]. 

EEG is not pathognomonic, but its alterations, commonly in the form of abnormal background activity, typically present occipital predilection [5,8,11]. Mitochondrial patients with a history of seizures have higher rates of brain MRI abnormalities (cortical/subcortical atrophy, stroke-like lesions, basal ganglia and white matter signal alterations) and increased serum lactic acid [5,7,8].

Patients with early-onset epilepsy, a history of tonic seizures with a tendency to recur and a disorganized background activity in EEG traces would primarily benefit from preliminary nuclear gene testing [8]. In this case, next-generation sequencing methodologies providing complete coverage of known nuclear mitochondrial genes (including *POLG* and LS-associated genes) is preferred, and single-gene testing should be avoided given the genetic pleiotropy and syndromic overlapping in primary mitochondrial diseases [80]. In the case of early-onset epilepsy, with seizures of unknown onset at the beginning with occurrence of epileptic spasm, testing for m.8993T > G (MILS mutation) should be provided firstly [7]. Patients with myoclonic seizures, especially when suffering from neuromuscular disorders (myopathy and ataxia), would more likely warrant diagnostic testing for m.8344A > G (MERRF mutation) [7,8]. Finally, in the presence of a history of status epilepticus or seizures associated with hearing loss, short stature, endocrine disorders or migraine, m.3243A > G (MELAS mutation) should be screened firstly [7,8,9]. In the case of a strong likelihood of primary mitochondrial disease and negative blood testing for the most common mtDNA point mutations, patients should have mtDNA assessed in another tissue, such as muscle. Especially in cases of MELAS suspicion, heteroplasmy analysis in urine can selectively be more informative [80].

## 7. Key Points

-Seizures and status epilepticus represent one of the most frequent symptoms of mitochondrial diseases; approximately 20–50% of mitochondrial patients will have seizures during their disease course.-Pathomechanism of mitochondrial epilepsy is not totally understood, but the role of mitochondrial defects in the development of seizures is well recognized.-Patients with specific genotypes are more at risk of showing seizures in their lives than others, and they need to be followed up on to prevent these events (m.3243A > G, m.8344A > G, *POLG*).-The management and treatment of mitochondrial epilepsy need to be personalized for each patient, due to the variability of phenotypes that mitochondrial patients may present.

## Figures and Tables

**Figure 1 ijms-23-13216-f001:**
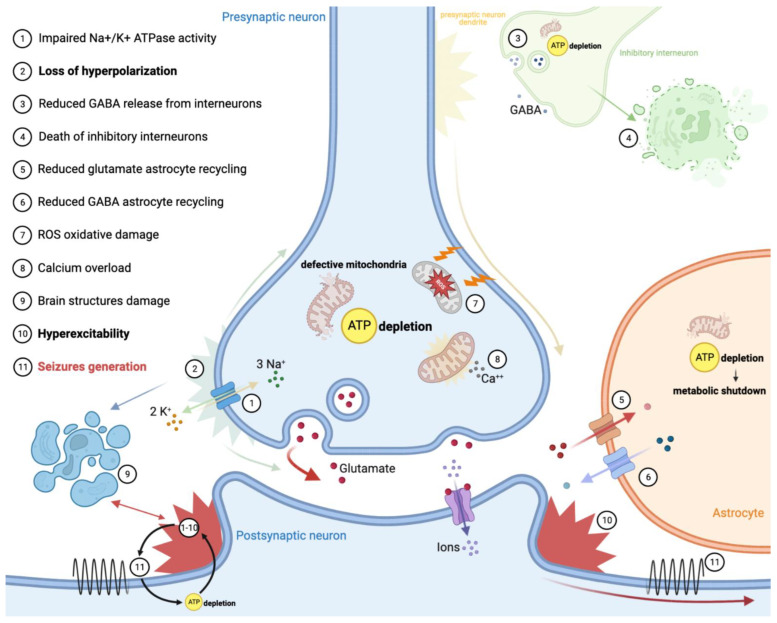
Epilepsy pathomechanism in primary mitochondrial diseases.

**Figure 2 ijms-23-13216-f002:**
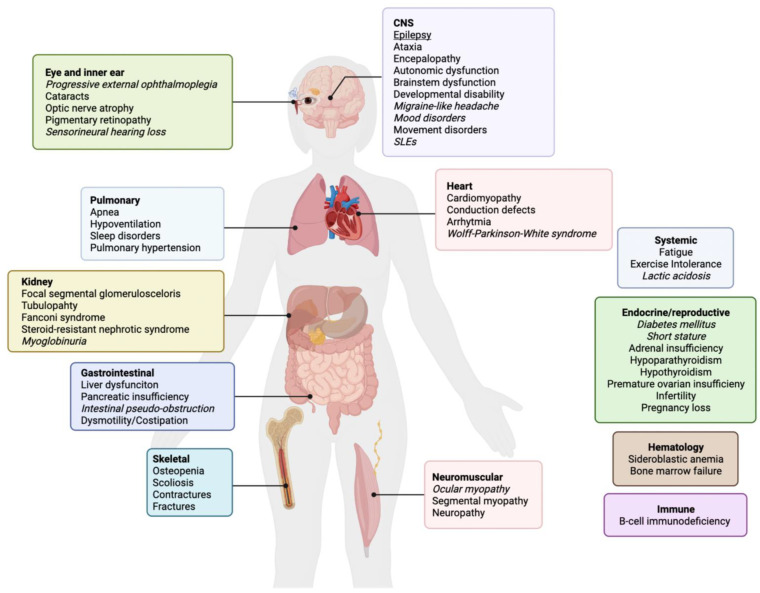
Phenotypic expression of primary mitochondrial diseases. In italics: peculiar signs/symptoms (‘red flags’) suggestive of mitochondrial disorder.

**Table 1 ijms-23-13216-t001:** Gene defects and phenotypes associated with mitochondrial epilepsy.

Phenotypes	Genes	References
MELAS	*MT-TL1* (3243A > G, 3271T > C), *MT-ND5* (m.13513G > A)	[14]
MERRF	*MT-TK* (m.8344A> G, 8356T > C, 8363G > A, 8361G > A)	[15,16]
*POLG-related disorders spectrum* (MCHS, AHS, MEMSA, ANS)	*POLG* (c.1399G > A, c.2243G > C, c.2542G > A)	[17,18]
Leigh syndrome	>75 nuclear-encoded genes	[19]
MILS	*MT-ATP6* (8993T > G)	[19,20]
Pyruvate dehydrogenase complex deficiency	*PDHA, PDHB, LIAS,* *LIPT1, DLD, PDH*	[21]
Others (Leigh-like syndrome, AHS, MDS, NAS encephalopathy)	*NDUFA13, ATP5A1, NDUFAF2, NDUFAF3, NDUFAF4,* *ACAD9, SCO2, FASTKD2, COX10, COX15, TMEM70, ANT1, TYMP, SUCLA2, DGUOK, RRM2B, FBXL4, TFAM, CARS2, DARS2, NARS2, PARS2, RARS2, VARS2, TARS2, TSFM, GTPBP3, RMND1, MRPL12, COQ2, COQ5, COQ8A, COQ9, ETHE1, ATAD3, SLC25A22, AIFM1*	[22,23,24,25,26,27,28,29,30,31,32,33,34,35,36,37,38,39,40,41,42,43,44,45,46,47,48,49,50,51,52,53,54,55,56]

MCHS: myocerebrohepatopathy, AHS: Alpers–Huttenlocher syndrome, MEMSA: myoclonus epilepsy myopathy sensory ataxia, ANS: ataxia neuropathy spectrum, MILS: maternal inherited Leigh syndrome, MDS: mtDNA depletion syndrome, NAS: not otherwise specified. The OMIM (https://www.omim.org) and MITOMAP databases (https://www.mitmap.org) were consulted on 28 September 2022.

## Data Availability

Not applicable.

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
