# Peer review of "Mitochondrial Epilepsy, a Challenge for Neurologists"

_ijms, 2022, doi:10.3390/ijms232113216_

Round 1

Reviewer 1 Report

The review paper “Mitochondrial epilepsy. A challenge for neurologists” approaches the epilepsy in primary mitochondrial diseases: prevalence, symptoms and pathophysiology. The paper also describes the most common mitochondrial diseases and the associated epilepsy phenotypes.

The last reviews about epilepsy in mitochondrial diseases were published in 2020 (in European Journal of Paediatric Neurology) and in 2021 (in Children). In my opinion, the major difference between this review and the previous reviews is the pathophysiological component and the underlying mechanisms. The authors should reinforce this part of the pathophysiology. The authors could provide more details about the mechanisms involved in the excitotoxicity. For example, how does defective mitochondrial metabolism increase glutamate release by astrocytes?

Overall, the paper is interesting and well-written, and the images are appealing. I have only two minor comments:

1)    recommend the authors to avoid the use of unnecessary abbreviations that hinders the reading. Examples of designations that do not need abbreviation: primary mitochondrial diseases or status epilepticus.

2)    Please improve the sentence in order to become more specific “More than 75 nuclear disease-causing monogenic mutations have been recognized” (page 7).

Author Response

Dear Reviewer,Thank you very much on the interest regarding our article. Your comments were very useful.

In detail:

1)    recommend the authors to avoid the use of unnecessary abbreviations that hinders the reading. Examples of designations that do not need abbreviation: primary mitochondrial diseases or status epilepticus.

- We removed the following abbreviations: CNS, AHS, MEMSA, SCAE, ANS, SLLs, SE, EPC, PMDs, KD, ASM. We retained the following major abbreviations: phenotypes (MERRF, MELAS, LS etc) and stroke-like episodes (SLEs).

2)    Please improve the sentence in order to become more specific “More than 75 nuclear disease-causing monogenic mutations have been recognized” (page 7).

  • We made more specific the description of LS genetic basis.

We all hope you will find the revised manuscript (chenges in bold) acceptable for publication.

Thank you very much and best regards

Michelangelo Mancuso

Reviewer 2 Report

This narrative review focuses and addresses the most recent advances in the field of mitochondrial epilepsy, discussing its pathophysiology, the different genetic etiologies, up to considering the currently available therapeutic treatment options.

I have no substantial criticisms to detect in the content of the manuscript, which is well written provideing an informative and detailed overview on this topic. The review in my opinion can be accepted in its current form.

Author Response

thank you. Michelangelo Mancuso

Reviewer 3 Report

The topic addressed in this review is very interesting, given the importance of the problem of epilepsy. The paper is well structured and the schematizations shown in figure 1 and figure 2 are very explanatory.

I suggest minor revision:

·        The introduction paragraph is unbalanced in size with respect to the following ones. In my opinion it should report more information on mitochondria, both from the biological point of view and, subsequently, from the pathological point of view. This would make the paper clearer for readers not working in this specific field.

·        More discussed should be the fact that Mitochondrial disorders concerns several heterogeneous clinical syndromes: encephalo- and / or cardiomyopathies.

·        Finally, some recent case reports reporting the relationship between some specific drug treatments and mitochondrial epilepsy should be reported and discussed.

Overall the paper is interesting and I am in favor for the  publication on IJMS journal.

Author Response

Dear Reviewer,

Thank you very much on the interest regarding our article. Your comments were very useful.

In answer to them:

  • The introduction paragraph is unbalanced in size with respect to the following ones. In my opinion it should report more information on mitochondria, both from the biological point of view and, subsequently, from the pathological point of view. This would make the paper clearer for readers not working in this specific field.
  • We added a brief introduction of mitochondrial biology in the introduction paragraph.
  • More discussed should be the fact that Mitochondrial disorders concerns several heterogeneous clinical syndromes: encephalo- and / or cardiomyopathies.
  • We added a brief statement about the spectrum of mitochondrial disease phenotypic expression in the introduction paragraph. The full list of the clinical features of primary mitochondrial diseases is shown in Figure2.
  • Finally, some recent case reports reporting the relationship between some specific drug treatments and mitochondrial epilepsy should be reported and discussed.
  • We added a 2020 paper concerning drug safety in primary mitochondrial disease and mentioned other anti-epileptic drugs. As far as we know, we have performed a detailed PUBMED search (last check October 20th), and no more outstanding articles to be discussed or cited are published. Finally, we made the pharmacological treatment paragraph more tidy.

We all hope you will find the revised manuscript (chenges in bold) acceptable for publication.

Thank you very much and best regards

Michelangelo Mancuso